

# Effects of *in situ* climate warming on monarch caterpillar (*Danaus plexippus*) development

Nathan P. Lemoine[1,4], Jillian N. Capdevielle[2] and John D. Parker[3]

[1] Department of Biological Sciences, Florida International University, Miami, FL, USA
[2] Department of Environmental Science, Policy, and Management,
University of California—Berkeley, Berkeley, CA, USA
[3] Smithsonian Environmental Research Center, Edgewater, MD, USA
[4] Current affiliation: Department of Biology, Colorado State University, Fort Collins, CO, USA

## ABSTRACT

Climate warming will fundamentally alter basic life history strategies of many ectothermic insects. In the lab, rising temperatures increase growth rates of lepidopteran larvae but also reduce final pupal mass and increase mortality. Using *in situ* field warming experiments on their natural host plants, we assessed the impact of climate warming on development of monarch (*Danaus plexippus*) larvae. Monarchs were reared on *Asclepias tuberosa* grown under 'Ambient' and 'Warmed' conditions. We quantified time to pupation, final pupal mass, and survivorship. Warming significantly decreased time to pupation, such that an increase of 1 °C corresponded to a 0.5 day decrease in pupation time. In contrast, survivorship and pupal mass were not affected by warming. Our results indicate that climate warming will speed the developmental rate of monarchs, influencing their ecological and evolutionary dynamics. However, the effects of climate warming on larval development in other monarch populations and at different times of year should be investigated.

## INTRODUCTION

Modified temperature regimes caused by climate change will fundamentally alter insect life cycles. As with other insects, lepidopteran larval development is temperature-dependent. Warming increases growth rates and survivorship; however both growth and survival decline rapidly once temperatures exceed an individual's thermal optimum (*Kingsolver et al., 2006*; *Kingsolver & Woods, 1997*). The effects of elevated temperatures on lepidopteran larval development have, to date, been mostly examined in highly controlled lab settings. Such laboratory experiments cannot incorporate natural temperature fluctuations that affect larval development and survival (*Zalucki, 1982*) or changes in insect behavior (i.e., behavioral thermoregulation, predator avoidance). Furthermore, warming alters plant nutritional quality (*Veteli et al., 2002*), and lab experiments often use artificial foods (*Kingsolver et al., 2006*; *Lee & Roh, 2010*) or leaf material that was not grown under elevated temperatures (*Lemoine, Burkepile & Parker, 2014*). Extrapolating results from laboratory experiments to natural settings is therefore problematic. Field experiments are necessary to

Corresponding author
Nathan P. Lemoine,
lemoine.nathan@gmail.com

identify how elevated temperatures influence insect development in a more natural, albeit, variable environment.

Monarch butterflies (*Danaus plexippus*) are a charismatic species found throughout North America and are well known for their annual migrations between Mexico and northern United States and southern Canada. Monarch migrations have been extensively studied, focusing on factors that influence migration success and population size (*Reppert, Gegear & Merlin, 2010*; *Flockhart et al., 2015*), potential overwintering and migratory habitat loss (*Oberhauser & Peterson, 2003*; *Pleasants & Oberhauser, 2012*; *Sáenz-Romero et al., 2012*), and overwintering behavior (*Masters, Malcolm & Brower, 1988*). Reductions in overwintering and migratory habitat caused by changes in climate and land-use have stimulated research on thermal constraints experienced by migratory adults and larvae, the need for cool night time temperatures to induce reproductive diapause in adult monarchs (*Goehring & Oberhauser, 2002*; *Guerra & Reppert, 2013*), and the threat posed by spring droughts that reduce monarch population sizes in their summer breeding grounds (*Zipkin et al., 2012*).

As with all insect species, monarch larval growth, consumption, and mortality rates depend upon environmental temperatures (*Zalucki, 1982*; *Goehring & Oberhauser, 2002*; *York & Oberhauser, 2002*; *Lemoine, Burkepile & Parker, 2014*). Prolonged exposure to extreme heat reduces larval growth and survival rates in laboratory experiments (*Zalucki, 1982*; *York & Oberhauser, 2002*). Although warming alters the nutritional quality of monarchs' milkweed host plants (*Couture, Serbin & Townsend, 2015*), few studies consider concurrent effects of warming on both monarch and milkweed (*but see Couture, Serbin & Townsend, 2015*). Milkweed nitrogen, lignin, and fiber content increase under elevated temperatures (*Couture, Serbin & Townsend, 2015*). Given that elevated temperatures affect both monarchs and milkweeds simultaneously, the relationship between temperature and monarch larval development and survival might be fundamentally different under climate warming. Field experiments are necessary to explore how warming affects monarch larval development in a scenario that incorporates natural temperature variability and changes in host plant quality.

Here, we report results from an *in situ* warming experiment designed to assess how elevated temperatures influence growth, survival, and development of monarch larvae under variable field conditions. We hypothesized that development time would decrease with rising temperatures under ambient conditions, as has commonly been reported for monarch larvae (*Zalucki, 1982*). However, this relationship between development time and temperature should be significantly stronger under warming since milkweed grown under elevated temperatures contains significantly more nitrogen (*Couture, Serbin & Townsend, 2015*). We expected that pupal mass and survival would decrease with rising temperatures (*Zalucki, 1982*; *York & Oberhauser, 2002*), but that warming would weaken these effects due to the effects of elevated temperatures on milkweed nutritional quality (*Couture, Serbin & Townsend, 2015*).

## METHODS

All experiments were conducted at the Smithsonian Environmental Research Center in Edgewater, MD. The experiment consisted of 16 replicate 2 × 2 m garden beds. Garden beds were assigned to temperature treatments in a completely randomized design. Warming treatments were imposed using a single Kalglo MRM-1215 1,500 W (Kalglo Electronics Company, Bethlehem, PA) heater installed 1.5 m from the ground over half of the garden beds. An aluminum frame of the same shape and size as the heaters was hung over the remaining garden beds to mimic any shading effects ($n = 8$ garden beds per temperature treatment). In each garden bed, 1 m long, 20 cm high aluminum sheets were driven 10 cm into the soil to quarter the 2 × 2 m garden bed into four 1 × 1 m subplots. In the fall of 2013, butterfly weed *Asclepias tuberosa* was sown into two of the subplots within each garden bed, resulting in a density of ∼4 plants per subplot (the remaining two subplots were used for other experiments). We chose *A. tuberosa* over the *A. syriaca* because *A. syriaca* can grow to >2 m tall, surpassing the height of our heaters. The experimental unit was therefore 32 1 × 1 m subplots ($n = 16$ per temperature) within the sixteen garden beds.

We placed Onset HOBO temperature loggers in the center of each garden bed to record air temperature in 10-minute intervals over the course of the experiment. In 'Ambient' treatments, average daytime temperatures were $25.2 \pm 1.4$ °C and average nighttime temperatures of $19.9 \pm 2.0$ °C. Maximum daytime temperatures at our study site were $30.7 \pm 2.5$ °C, while minimum nighttime temperatures were $18.2 \pm 2.3$ °C. Since air temperature measurements may not accurately reflect the heating achieved by infrared heaters (*Kimball et al., 2008*), we verified heating treatments using a handheld IR thermometer to measure temperatures on a plastic sphere mounted 0.5 m from the ground placed in the middle each experimental subplot at midnight. Nighttime IR gun measurements verified that heaters raised surface temperatures by ∼4 °C on average ($p < 0.001$), which is below severe projections of a 6 °C increase in temperature but above the more conservative estimate of a 2 °C temperature increase by 2100 (*IPCC, 2007*).

In August 2014, monarch eggs and larvae were gathered from *A. syriaca* within nearby old-growth fields. Eggs and larvae were reared in mesh cages and fed fresh *A. syriaca* leaves daily until they reached the third instar. Larval development was checked continuously throughout the day. First or second instars escaped the mesh bags easily and thus were not used. Immediately after molting to the third instar, larvae were randomly assigned to a temperature treatment ('Ambient', 'Warmed') and a single larva was placed on a single *A. tuberosa* within a randomly chosen plot ($n = 15$, $n = 18$ for 'Ambient' and 'Warmed' treatments, respectively). A 20 × 30.5 cm organza mesh bag was placed over the plant to retain the monarch. If the monarch larva consumed the entire host plant, they were transferred to another plant within the same subplot. Time to pupation was recorded as the number of hours between experiment initiation and onset of chrysalis formation, and this number was converted to number of days (development hour/24). Dead individuals were recorded and removed from the host plant. Chrysalids were transported to the lab and weighed to obtain final pupal mass.

We measured three plant traits (specific leaf area (SLA), water content, and latex production) to determine whether warming effects on monarch development might be mediated through warming effects on plant traits. At the end of the experiment, two newly expanded leaves were collected from each plant. For one leaf, we measured leaf area, obtained a fresh wet mass, and then dried the leaf to obtain a dry mass. We calculated specific leaf area (SLA) as area/dry mass and percent water content as $(1 - \text{dry mass(g)}/\text{fresh mass (g)}) * 100$. Using the second leaf, we determined latex production by cutting the tip of the leaf and blotting all latex onto a dry, pre-weighed piece of filter paper (*Agrawal, 2005*). The filter paper was dried again and latex concentration calculated as the difference in post- and pre-latex filter weights divided by leaf area (*Agrawal, 2005*).

Although heaters raised temperatures of 'Warmed' plots by ~4 °C on average, plots varied considerably in temperature due to different light levels across the experimental garden and varying plant biomass within each plot. We therefore measured temperature with a handheld infrared thermometer in each subplot during the night at the end of the experiment. For consistency, we recorded temperature of a white plastic sphere mounted 0.5 m from the ground in the middle of each subplot. We then treated temperature as a quantitative rather than categorical variable in all analyses. Note that these measures reflect relative differences in temperature among plots that should be relatively constant over the experiment.

We used an ANCOVA design to regress days$^{-1}$ until pupation and final pupal mass against night-time temperatures as measured by the IR gun. We included temperature treatment as a covariate, which allows for the possibility that slopes differ between temperature treatments. Mortality was assessed using logistic regression that also included night-time temperature and its interaction with temperature treatment, as in the ANCOVA. Although monarchs experience mortality as pupae, brief exposure to prolonged temperatures did not alter pupal mortality rates and third instars were the most sensitive to temperature increases (*York & Oberhauser, 2002*). Thus, our experiment likely captured most of the influence of temperature on larval survival.

Model assumptions were verified with residual plots where appropriate. All analyses were conducted using Python v2.7 with the '*numpy*', '*pandas*', and '*statsmodels*' modules (*McKinney, 2010*; *Seabold & Perktold, 2010*; *Van der Walt, Colbert & Varoquaux, 2011*).

## RESULTS

Time to pupation decreased with increasing temperature, but did so differently under 'Ambient' and 'Warmed' conditions (interaction: $p < 0.041$) (Fig. 1). At the lowest temperature in 'Ambient' treatments, 12.6 °C, monarch larvae required 12.2 days to transition between third instar and pupa. At the warmest temperature achieved in the 'Warmed' plots, monarch larvae required only 10.0 days to pupate. Importantly, the relationship between temperature and time to development varied among treatments (Fig. 1). When reared under 'Ambient' conditions, larval development time decreased by ~0.4 days per 1 °C increase in temperature. In 'Warmed' plots, larval development time decreased by

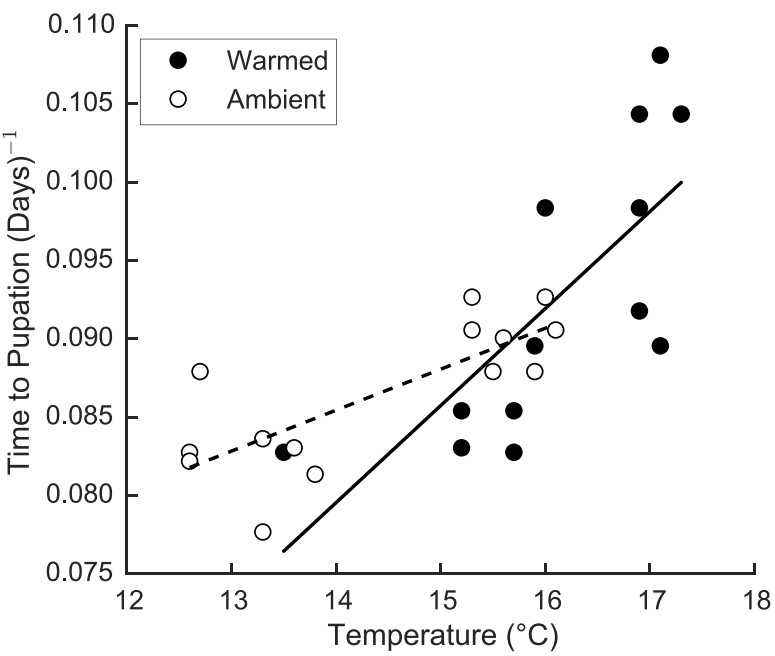

**Figure 1 Effects of temperate on monarch development time.** Monarch development time decreased as temperature increased in both 'Ambient' (open circles) and 'Warmed' (filled circles) plots. However, the effect of temperature on monarch larval development was stronger in 'Warmed' plots.

~0.7 days per 1 °C increase in temperature. Climate change may therefore speed larval development by ~0.7–2.4 days, depending on the severity of temperature increases.

Air temperature measurements do not accurately reflect the intensity of infrared heating because infrared energy warms surfaces and not the air (*Kimball et al., 2008*), calculations of degree-days may not accurately reflect the underlying temperature treatments. Still, we calculated the number of degree days experienced by each individual for which there was adequate temperature data following the simple averaging method, since temperatures remained within the upper and lower thermal limits throughout the experiment (*Allen, 2006*) . Monarch caterpillars experienced ~155 ± 17 degree days, and this did not differ between temperature treatments ($p = 0.978$). Thus, monarch larvae accumulated their required number of degree days faster in the warming treatment than in the ambient treatment.

Temperature had no effect on pupal mass ($p = 0.454$, $R^2 = 0.023$). Similarly, mortality was low throughout the experiment (18%) and independent of temperature ($p = 0.610$, pseudo-$R^2 = 0.01$).

Warming had no effect on any measured plant trait. SLA ($p = 0.940$, $R^2 = 0$), percentage water content ($p = 0.313$, $R^2 = 0.05$), and latex concentration ($p = 0.739$, $R^2 = 0.01$) all did not vary with temperature. Thus, any effects of warming on monarch development time were direct effects of temperature on monarch physiology rather than being mediated through the plant traits we measured.

## DISCUSSION

Our study indicates that warming accelerates monarch larval development but has little effect on larval mortality or pupal mass at our study site. This is consistent with numerous studies showing positive correlations between larval development and temperature (*Kingsolver & Woods, 1997*; *Bale et al., 2002*). Since warming increases larval growth rates, lepidopteran larvae reach critical mass needed for pupation earlier and proceed through larval stadia more quickly. Monarch larvae developed more rapidly from the third instar but experienced roughly the same number of degree days. Our results suggest that climate warming might facilitate monarch larval development through later instars under moderate climate change scenarios at sites with relatively cool temperatures, potentially increasing the number of generations in the temperate summer breeding grounds of eastern migratory monarch populations.

Laboratory studies have consistently documented negative effects of extreme temperatures on monarch caterpillar development and survival. Short-term, extreme heat stress can have weak negative effects on pupal mass (*York & Oberhauser, 2002*). Likewise, constant temperatures above 28 °C induced high mortality rates in monarch larvae (*Zalucki, 1982*; *York & Oberhauser, 2002*). However, these studies used either pulses of extremely high temperatures (i.e., 36 °C) or held monarch larvae at a constant temperature (i.e., 28 °C). Ambient, maximum daytime temperatures averaged 30 °C during our experiment; warming increased this maximum to 32–34 °C. Although these temperatures are above the thermal optimum of monarch survival, we found no effect of *in situ* warming on either pupal mass or survival of older monarch larvae. Eggs and first instar larvae are resistant to high temperatures, with third instars, fourth instars, and pupae being the most sensitive to extreme heat (*Zalucki, 1982*; *York & Oberhauser, 2002*). As temperatures exceeded 28 °C for less than 20% of the full 24 h day, it is likely that diel and daily temperature fluctuations mitigated the lethality of high temperatures.

Interestingly, our study site had warmer temperatures during our experiment than other locations of the monarch breeding range. Monarchs typically experience cool temperatures during their northward migration: maximum March temperatures in Texas average $23.5 \pm 2.4$ °C, maximum April temperatures in Iowa and the midwestern US average $20.7 \pm 1.5$ °C, and maximum May temperatures in the Great Lakes region average $18 \pm 2.3$ °C (averages based on 50 year weather station data provided by WorldClim). Even maximum temperatures during the summer breeding season in the Great Lakes region are typically lower than at our study site, averaging $26.0 \pm 2.3$ °C compared to $30.7 \pm 2.5$ °C at during our experiment. We found no influence of increased temperatures on larval monarch pupal mass and survival at our study site, which had temperatures well above those in other important breeding ranges. Indeed, temperatures in these ranges rarely exceed the thermal optimum of 28 °C and do not exceed the critical thermal maximum of 36 °C (*Zalucki, 1982*; *York & Oberhauser, 2002*). Thus, climate change is unlikely to raise temperatures to a range that is detrimental to monarch larval performance.

Though monarch larval development proceeded more rapidly when exposed to high temperatures, this effect was stronger on plants grown under warmed conditions.

Increased sensitivity to temperature in 'Warmed' plots likely stems from altered plant nutritional content. Though we found no different in *A. tuberosa* leaf characteristics between 'Ambient' and 'Warmed' treatments, elevated temperatures alter foliar water content, nutritional content, and secondary metabolite concentrations of numerous plant species (*Zvereva & Kozlov, 2006*). In particular, milkweed nitrogen content increases at elevated temperatures (*Couture, Serbin & Townsend, 2015*). Insect development proceeds more rapidly at high temperatures on nitrogen-rich plants (*Lemoine et al., 2013*; *Lemoine, Burkepile & Parker, 2014*). It is therefore likely that increased foliar nitrogen content of *A. tuberosa* grown under elevated temperatures is responsible for the greater sensitivity of monarch larvae to rising temperatures.

Our paper demonstrates that climate warming may minimally impact the development of monarch larvae in temperate regions. Though numerous laboratory studies have reported detrimental impacts of extreme temperatures on monarch larval development and survival (*Zalucki, 1982*; *York & Oberhauser, 2002*), our field experiment demonstrated that *in situ* warming had little influence on larval survival or pupal mass even in a site with extreme daytime temperatures. Rising temperatures may, however, have other important effects on monarch larvae. Monarch larvae may, for example, suffer higher parasitism rates at high temperatures as occurs in other insect-parasitoid pairs (*Bezemer, Jones & Knight, 1998*). Predatory insects also increase their attack and ingestion rates at high temperatures, suggesting that predation pressure on monarch larvae may increase substantially under warming (*Rall et al., 2010*). Furthermore, landscape level distributions of milkweed host plants may be substantially different at elevated temperatures. Warming may reduce the availability of *Asclepias* host plants during the northward migration via increased drought or drastically alter the geographic range of *Asclepias* host plants (*Zipkin et al., 2012*; *Lemoine, 2015*). Thus, this research establishes an important baseline for future work considering numerous other consequences of increased temperature on monarch larval performance and survival.

## ACKNOWLEDGEMENTS

We thank S Cook-Patton, D Doublet, and M Palmer for their assistance during this project. We thank L Higley, J Pleasants, R Peterson, and one anonymous reviewer for their helpful comments and suggestions.

### Funding

This work was funded by an NSF DDIG (DEB-1311464), a Smithsonian Graduate Research Fellowship, and an FIU DYF to NPL and an NSF REU grant (DBI-156799) to JDP. The funders had no role in study design, data collection and analysis, decision to publish, or preparation of the manuscript.

## Grant Disclosures

The following grant information was disclosed by the authors:
NSF DDIG: DEB-1311464.
Smithsonian Graduate Research Fellowship.
FIU DYF.
NSF REU: DBI-156799.

## Competing Interests

The authors declare there are no competing interests.

## Author Contributions

- Nathan P. Lemoine conceived and designed the experiments, analyzed the data, contributed reagents/materials/analysis tools, wrote the paper, prepared figures and/or tables, reviewed drafts of the paper.
- Jillian N. Capdevielle conceived and designed the experiments, performed the experiments.
- John D. Parker contributed reagents/materials/analysis tools, wrote the paper, reviewed drafts of the paper.

## Data Availability

www.natelemoine.com/data/.

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
