# Peer review of "Effects of in situ climate warming on monarch caterpillar (Danaus plexippus) development"

_PeerJ, doi:10.7717/peerj.1293_

## Round 0.1 · original submission · Major Revisions

· Academic Editor

Major Revisions

Drs. Lemoine, Capdevielle, and Parker,

I find myself in the somewhat unusual position of being rather more critical of your manuscript than the reviewers. All of us (the reviewers and I) agree that your manuscript would be acceptable with revision, although we differ on the specifics of the revision. The key point of disagreement I have with the reviewers is that while they found the analysis acceptable, I did not. As I state in the notes in your manuscript, I think you are doing yourselves a disservice by not conducting separate regressions by treatment. In any event, please look at the reviewers comments and at my notes on an annotated pdf of your submission. I should mention that although reviewer one finds your study to be unreplicated, I believe the actual point is that only one year's environmental data are provided. I don't believe there is a fast rule on when entire experiments need to be repeated (I prefer that term to avoid confusion with replication in an experiment), but I don't see the need in this case given that the treatments (supplemental warming or nothing) were so simple and responses relatively straightforward.

Other than the analysis issue, the only points of great substance are the need for substantial revision of the abstract and introduction (ample suggestions are provided by the reviewers), and the point made by reviewer one regarding this not being a "climate change" manuscript. Please consider those comments, and I recommend being conservative in your choice of terms.

In your rebuttal letter, please be clear regarding choices you made in making or not making changes. And please feel free to contact me during the revision process if I can be of any assistance. Although the following (boiler-plate) paragraphs mention the usual policy of re-review following major revisions, I am hopeful this won't be necessary.

·

Basic reporting

The manuscript provides information on the effect of temperature on development, biomass, and mortality of monarch butterfly larvae. The paper is well written, but there are several items the authors must address.

Experimental design

There is only one replication of the experiment. This is potentially problematic because this field study involves insects, plants, insect feeding on plants, and environmental factors. How can the authors and readers be confident that the results observed were not the result of unique environmental circumstances in 2014? Repeating the experiment in time (or at the very least in space) would lead to much more confidence in the results. The authors capture third instars on Asclepias syriaca in an old field and place them on A. tuberosa. The authors need to provide much more information on the experimental and treatment design. How can the authors and readers be confident that the results observed were not caused by an interaction between temperature and larvae moving from A. syriaca to A. tuberosa? What did the authors do to control for possible cage effects? There should have been plots without cages as an extra control. This is especially important given that cages are well known to alter the microclimate of plants and the subject of the present study is focused on effects of environment on the plants, monarch larvae, and the interactions between the two. What was the seeding rate? What were the final plant counts per subplot? Were the plants watered or rain-fed only? If they were watered, how did you deal with the fact that the warmed plants required more water because of greater metabolism, transpiration rates, etc.? How many larvae were placed on each plant in a subplot? What about larval density effects? Was it a completely random design with paired treatments or randomized complete block design with split plots? Based on the reported results, seems like natural light could have been used as a blocking factor. What was the other half of each plot used for? The authors mention another study, but provide no details. Each experimental unit is a split-plot with a pairing of control (ambient temp) and elevated temp, but it seems the authors treat each observation as independent in their regression analysis. This may not be the appropriate analysis, although I believe some type of regression analysis is superior to their initial analysis plans.

Validity of the findings

A major issue is that the title and almost the entire focus of the paper is on climate warming. Although liberties can be taken when writing grant proposals, the reporting of studies needs to avoid speculation and stick to what was actually manipulated and measured in the experiments. This study is not about climate warming. The authors did not manipulate climate and did not assess “the impact of climate warming on development of monarch…” Rather, they examined the influence of temperature on development and other biological attributes of larvae, pupae, and A. tuberosa. Climate is not simply temperature over the fall of one year (2013) and the growing season of the next year (2014). Consequently, the paper needs to be rewritten with a focus on how elevated temperatures in an outdoor environment affected larval development. The authors can then engage in some mild speculation in the discussion section about how their results might be extrapolated to climate change.

Another issue is the lack of assessment of eggs, neonates, and second instars. It is well known that many (most?) lepidopteran species experience the majority of mortality as neonates and about 90-99% mortality from egg to adult. By only using third through pupal stages in the present study, the results may not adequately reflect mortality or biomass differences between ambient and warmed environments.

It is interesting that the study showed a lack of mortality and biomass effects (but see above paragraph) whereas other studies have shown effects. The authors suggest that this could be because of the outdoor environment and fluctuating diurnal temperatures in their study versus the constant temperatures in growth chamber experiments of previous studies. However, this was not specifically tested in the present study, so not much more can be said.

If the paper is restructured so that it focuses on temperature rather than climate warming and the mortality and biomass results are considerate of the fact that eggs, neonates, and second instars were not included in the study, then what is left are results that show that late larval development is temperature-dependent. This is a first principle of poikilotherm biology, so it could be argued that the results are trivial. The possible extenuating circumstances here are that the authors have demonstrated this in an outdoor environment with fluctuating temperatures. However, the authors may want to consider not publishing this information as a separate paper, but rather combining the data from this paper with other data sets to relate a more comprehensive story.

Specific Comments:
Line 1. Change title to accurately reflect the scope of the study. See above.
L 27. Delete first sentence.
L 31. On first use, write out the ESA approved common name of the species, i.e., monarch butterfly. Then, indicate that “hereafter referred to as “monarch caterpillar” “monarch larva” etc. Or, just write monarch butterfly larvae each time.
L 31. Include the common name of Asclepias tuberosa.
L 42. Change “considerably” to “considerable”.
L 71. Delete “in”.
L 86. Include a photograph or diagram of your experimental unit and experimental design.
L 91. Why did you use A. tuberosa? Why not A. syriaca?
L 91. What was the seeding rate? What were the final plant counts per subplot?
L 99. Were the plants watered or rain-fed only? If they were watered, how did you deal with the fact that the warmed plants required more water because of greater metabolism, transpiration rates, etc.?
L 100-107. This is a very confusing paragraph. Include a table of the relevant temperature statistics for the two treatments: ambient and warmed. Where were the data loggers placed within each plot?
L 108. Need to discuss in the discussion section potential limitations of collecting eggs and larvae from syriaca and transferring to tuberosa.
L 112. How many larvae were placed on each plant in a subplot?
L 113. These numbers do not match line 88’s comment about 16 plots. Very confusing.
L 114. Need dimensions, fabric type, and manufacturer for the mesh bags. A photograph would help.
L 119. Delete “carefully” and “back”.
L 120. How did you deal with different densities of larvae per plot?
L 136-7. Delete commas.
L 140. Spell out and define “OLS”.
L 143. Change “instar individuals” to “instars”. An instar is an individual.
L 169. Change “percent” to “percentage”.
L 175. See comments above about refocusing the paper.
L 219. Partial generations because of increased temperature may also be a problem if the population doesn’t “finish” the season as primarily adults that can then migrate.
L 220. Change “further” to “farther”.
L 227. Delete “in order”.
Figure 1. Include the relevant equation and statistics.
Figures 2, 3. Delete.

Reviewer 2 ·

Basic reporting

Abstract:
The abstract should heavily edited. The first sentence, although true, runs contradictory to your results. Lines 33 and 36 are redundant. I think that bringing up the contradiction to lab based studies should be brought up in the second to last sentence not the second. This point highlights why this study was necessary. I would omit the last sentence and replace it with a comment that highlights the importance of studying the impacts of Global Warming. Discussing future directions of this research should be mentioned in the discussion.

Introduction
Lines 41-44: Possible rewrite: "Modified temperature regimes caused by climate change may fundamentally alter insect 42 life cycles. As with other insects, lepidopteran larva development is temperature-dependent. Warming increases growth rates and survivorship; however yet growth and survival decline rapidly once temperatures exceed an individual’s …"

Line 48: development and survival (Zalucki 1982), which may not accurately reflect real climate

Lines 62-67: Possible rewrite: "Climate change may have considerable negative effects on monarch populations by reducing available overwintering and migratory habitat. In response, research has shifted to focus on thermal constraints experienced by migratory monarchs, development, cool night time temperatures inducing reproductive diapause in adult monarchs (Goehring and Oberhauser 2002, Guerra and Reppert 2013), and spring droughts reduce monarch population sizes in their summer breeding grounds (Zipkin et al. 2012)."

Line 73: Delete “However”

Line 77: Delete “Indeed”

Line 114: Possible rewrite: Does the reference to using first or second instars (should avoid writing “instar larvae” as that is repetitive) correspond to line 111 where you indicate that those molting to third instar were used? If so move the line “First or second instars escaped the mesh bags easily…”

Line 128: Is the determination of latex production based on a particular protocol used in other places?

Line 179: It seems like there should be more recent references here. You do say “numerous studies”. Is there a review article you could cite here that summarizes the large body of research on insect development related to temperature?

Line 181: “This is demonstrated by the fact that..” Poor word choice.

Line 195: how does diet mitigate the lethality of high temperatures?

Lines 197-206: The authors don’t do a very good job of tying this paragraph to your results. I think it does present important information but it seems to be stuck in as an afterthought. What does it mean that this study site represents the upper thermal limits? Expand on this.

Lines 207-208 “Climate change as been found to alter foliar water content, nutritional quality, and secondary metabolite concentrations (Zvereva and Kozlov 2006, Couture et al. 2015).” I’m not sure if they are referring to plant physiology as a whole or specifically milkweed plants. I know that there are studies that support this statement for plants as a whole but what body of literature supports this specifically for Asclepias?

207-213: This paragraph needs to be rewritten and the references need to be sorted out to make your key points clearer. Watch the overuse of conjunctions throughout the manuscript. They are distracting and too conversational.

Lines 214-227: I’m not sure the final paragraph is well served. The authors seem to disregard the importance of their study. Take the time to indicate that it is important that you established this baseline and that future research should focus on other implications on climate change and that your study is significantly different than research conducted in lab-based settings. Again, this is one of the important key points of this study. The final paragraph should stick to discussing implications related specifically to temperature increases rather than pointing out the broader impacts of climate change. For example, is there any evidence that higher temperatures facilitate parasitism or predation? I think one very important thing the authors do not mention in this paper is the implications climate change on the range of the milkweed plant itself. Climate change will rapidly change preferred growing temperatures for many host plants. This is mentioned but they do not discuss the problems that habitat disruption and availability present. Before drastic landscape changes caused by humans’ global changes in climate were overcome by the ability of plants and animals to change their ranges relatively rapidly. Because we are now dealing with a dramatically altered landscape there has been a loss of ecological plasticity.

Experimental design

This was an interesting experimental design. I appreciate the level of description.

Validity of the findings

In looking at figures 2 and 3 I find myself a bit confused about the data. They say that they do not see a significant difference between plots that were warmed and not warmed. That may be true but I think the authors need to run a trend line to make this clear to the reader. It does appear that the points are beginning to cluster and show some sort of trend at 16 C. In the results they report that mortality was low throughout the experiment and that it was independent of temperature. Why is almost all mortality associated with the warmed data points? The authors may want to explain that. I do not understand figure 3. Why present it as a probability at all? Perhaps this figure is unnecessary? I encourage the authors to reconsider how the data in figures 2 and 3 are being analyzed and/or presented.

Additional comments

This was an ambitious study. I recognize the difficulty of looking at the impact of temperature on organisms in the field. While these results did not show a negative impact on development I do think that the authors need to connect their findings to the larger body of literature on insect development. The authors really need to highlight that their field results are different than what is being found in lab-based settings and why this is important to the community studying the impacts of climate change as a whole.

·

Basic reporting

No comment

Experimental design

Appropriate experimental design although the sample size is small.

Validity of the findings

No major new ground is broken here but the conclusions are appropriate for the results found.

Additional comments

This paper reports on a study to examine the effect of temperature on monarch larval development. Although the increase in developmental time with temperature is well known this study performs an "in situ" experiment as opposed to previous studies which have been done in the lab. The results do show an increase in development time. Although the sample sizes are small, grouping "ambient" and "warmed" treatments together and treating measured temperature as a quantitative variable provides sufficient statistical power.
A few general comments: nighttime surface temperatures were used, presumably to get at how warm larvae would have gotten during the day. Was it not possible to take periodic surface temperature samples during the day? And at what time of night were temperature samples taken? Wouldn't that make a difference as cooling would occur during the nighttime?
Some editorial comments:
line 42: should be "considerable"
line 48 delete "which"
lines 54-55 delete "while ..." to end of sentence
line 57 the summer range of monarchs is not just the Great Lakes
Last paragraph should be shortened; already mentioned in Introduction.
Figs 2 and 3 can be eliminated; the stats in the text suffice.

---

## Round 0.2 · accepted · Accept

· Academic Editor

Accept

Thank you for your careful attention to the reviewers and my comments. I think the revised manuscript is clearer and better presents your findings. Consequently, I'm happy to accept it for publication. In reading the final version I did find one minor error on line 108 which was singular when it should have been plural, so I'm adding a note for the staff to make this change in the final publication (the sentence in question should start "The experimental units were..."

I do apologize for taking more days than I would have liked in getting back to you. Your revision came in just as I was leaving for Brazil (I'm teaching a short course on professional development and scientifc ethics), and this morning has been my first opportunity to meet my editorial responsibilities.